# Structural basis of proton-coupled potassium transport in the KUP family

Igor Tascón [1,5], Joana S. Sousa[2,5], Robin A. Corey[3], Deryck J. Mills[2], David Griwatz [1], Nadine Aumüller[1], Vedrana Mikusevic[1], Phillip J. Stansfeld [3,4], Janet Vonck[2]* & Inga Hänelt [1]*

Potassium homeostasis is vital for all organisms, but is challenging in single-celled organisms like bacteria and yeast and immobile organisms like plants that constantly need to adapt to changing external conditions. KUP transporters facilitate potassium uptake by the co-transport of protons. Here, we uncover the molecular basis for transport in this widely distributed family. We identify the potassium importer KimA from *Bacillus subtilis* as a member of the KUP family, demonstrate that it functions as a $K^+/H^+$ symporter and report a 3.7 Å cryo-EM structure of the KimA homodimer in an inward-occluded, trans-inhibited conformation. By introducing point mutations, we identify key residues for potassium and proton binding, which are conserved among other KUP proteins.

[1] Institute of Biochemistry, Goethe University Frankfurt, Frankfurt am Main, Germany. [2] Department of Structural Biology, Max Planck Institute of Biophysics, Frankfurt am Main, Germany. [3] Department of Biochemistry, University of Oxford, Oxford, UK. [4] School of Life Sciences & Department of Chemistry, University of Warwick, Coventry CV4 7AL, UK. [5] These authors contributed equally: Igor Tascón, Joana S. Sousa. *email: janet.vonck@biophys.mpg.de; Haenelt@biochem.uni-frankfurt.de

K$^+$ is the most abundant intracellular cation in all living organisms and its homoeostasis is absolutely essential. Microorganisms depend on a variety of different K$^+$ uptake systems to adapt to rapidly changing external conditions. The main players here can be grouped in two families, the Trk/Ktr/HKT/Kdp family[1] and the Kup/HAK/KT (K$^+$ Uptake, KUP) family[2–4]. Interestingly, members of both families are absent in mammalian cells, while they were identified as virulence factors in pathogenic bacteria[5–8]. The KUP family belongs to the APC (amino acid-polyamine-organocation) superfamily, the second largest superfamily of secondary active transporters[9]. Members of the KUP family have different physiological roles in diverse bacteria and even in different plant organs, with correspondingly a wide scope of affinities for K$^+$, ranging from nano- to millimolar concentrations[10–13]. KUP proteins have been proposed to be particularly important under acidic environmental conditions and to function as K$^+$/H$^+$ symporters[14–16]. Reporter fusions and cysteine labelling assays suggested the presence of 12 transmembrane helices and several residues crucial for K$^+$ transport activity have been identified in different KUP members[11,17–20]. Yet, due to the lack of detailed structural information KUPs' transport mechanism remains largely unknown.

Recently, KimA from *Bacillus subtilis* was identified as a high-affinity potassium importer[21] and KimA homologs were proposed to form a family within the APC superfamily. Here, we show that KimA is in reality a member of the KUP family. We functionally characterized KimA in vivo and in vitro and solved a trans-inhibited, inward-occluded structure using cryo-EM. In combination with mutational analysis, we propose a coupling mechanism for the symport of potassium ions and protons.

## Results

### KimA functions as a K$^+$/H$^+$ symporter

KimA was previously identified as a potassium uptake system in *Bacillus subtilis* cells depleted of its long-known potassium-importing channels, KtrAB and KtrCD[21], and is wide-spread among bacteria (Supplementary Fig. 1). To characterize the K$^+$ uptake mode of KimA, we performed in vivo and in vitro transport assays. In vivo, KimA mediates potassium uptake into potassium-depleted *Escherichia coli* LB2003 cells deficient in endogenous potassium uptake systems, with a $K_M$ value of 215 μM and a $V_{max}$ of 245 nmol min$^{-1}$ mg$^{-1}$ (Fig. 1a, b). Potassium uptake increased at lower external pH and was abolished in the presence of the proton ionophore CCCP (Fig. 1c). The observed pH dependency and the apparent requirement for a proton gradient suggest that KimA functions as a K$^+$/H$^+$ symporter. In vitro transport assays performed in proteoliposomes, in the presence of an inward-directed potassium gradient, confirmed that KimA couples potassium uptake and proton transport (Fig. 1d).

### Structure of dimeric KimA in inward-occluded conformation

To elucidate the structural basis for proton-coupled potassium transport, we determined a 3.7 Å resolution cryo-EM structure of KimA solubilized with styrene-maleic acid (SMA) co-polymers in the presence of potassium (Fig. 2, Supplementary Figs. 2 and 3 and Supplementary Table 1). Our structure shows that KimA forms a homodimer, with each protomer consisting of an N-terminal transmembrane (TM) domain and a C-terminal cytoplasmic domain (Fig. 2a, b), as previously suggested[22]. The dimer is stabilized by the swapping of the cytoplasmic domains, which are placed under the opposite membrane domains via long loops connected to the last TM helix (Fig. 2a). The interaction of these loops with the other cytoplasmic domain forms the main dimer interface (Fig. 2c). The two TM domains are tilted towards each other at the extracellular side, creating a second interface, and appear to enforce a bending of the membrane by ~130° against its natural curvature (Supplementary Fig. 4a). The same arrangement is observed for detergent-solubilized KimA (Supplementary Fig. 4b and c), demonstrating that this shape does not derive from forces exerted by the SMA co-polymer over the protein. However, we could not exclude that the depletion of lipids from the dimer interspace during purification may have led to the observed interface. To investigate this further, we carried out molecular dynamics (MD) simulations on the KimA dimer. An initial 100 ns coarse-grained (CG)[23,24] simulation on the positionally-restrained KimA dimer reveals that the membrane freely forms around the dimer (Fig. 3a; 'input'). This results in a substantial bending of the bilayer, in a similar arrangement to the one observed in the EM map. When investigating the dynamics of the unrestrained dimer over a total of 20 μs, we see that it relaxes into an 'upright dimer' arrangement with the TM domains separated (Fig. 3a). The dimer switches between both positions in a highly dynamic fashion (Supplementary Fig. 5a), with each arrangement about equally likely. Plotting the angle of each TM domain over the course of the simulation reveals that the two monomers move in concert, akin to a 'breathing' motion (Fig. 3b, Supplementary Fig. 5b, c). Such 'breathing' motion was also observed in the EM sample during the refinement, albeit to a lesser extent. Upright dimers were not observed in the cryo-EM analysis. Follow-up simulations totalling 6 μs using the atomistic CHARMM36 force field confirm the presence of the upright dimer, with 2/3 systems adopting the same pose (Fig. 3c and Supplementary Fig. 5d) and principal component analysis (PCA) on the data revealing that the breathing motion accounts for 68% of the total variance (Supplementary Fig. 5e). These MD simulations reveal that

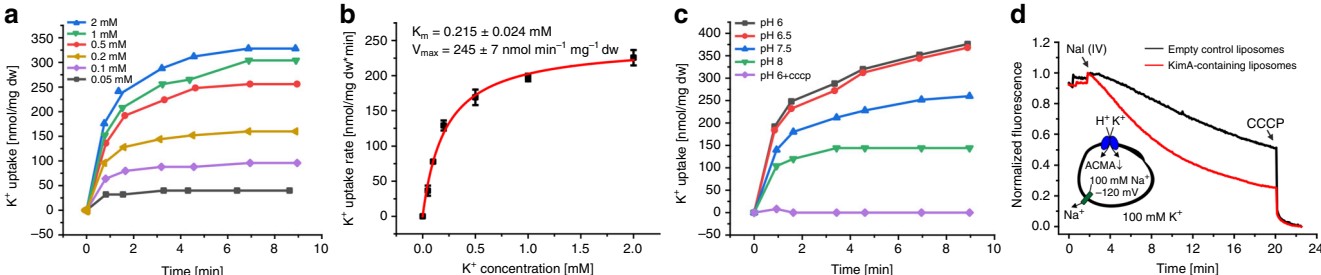

**Fig. 1 Functional analysis of KimA. a** Concentration-dependent potassium uptake via KimA into potassium-depleted *E. coli* LB2003 cells. $n = 3$ independent experiments; a representative experiment is shown. **b**, Kinetic parameters of the potassium transport via KimA determined based on (**a**). The plotted uptake rates at different potassium concentrations are the means of three independent experiments; errors shown are s.d. **c** pH-dependent potassium transport via KimA into potassium-depleted *E. coli* LB2003. $n = 3$ independent experiments; a representative experiment is shown. **d** Potassium-dependent proton transport into KimA-containing liposomes. $n = 3$ independent experiments; a representative experiment is shown. Source data are provided as a Source Data file.

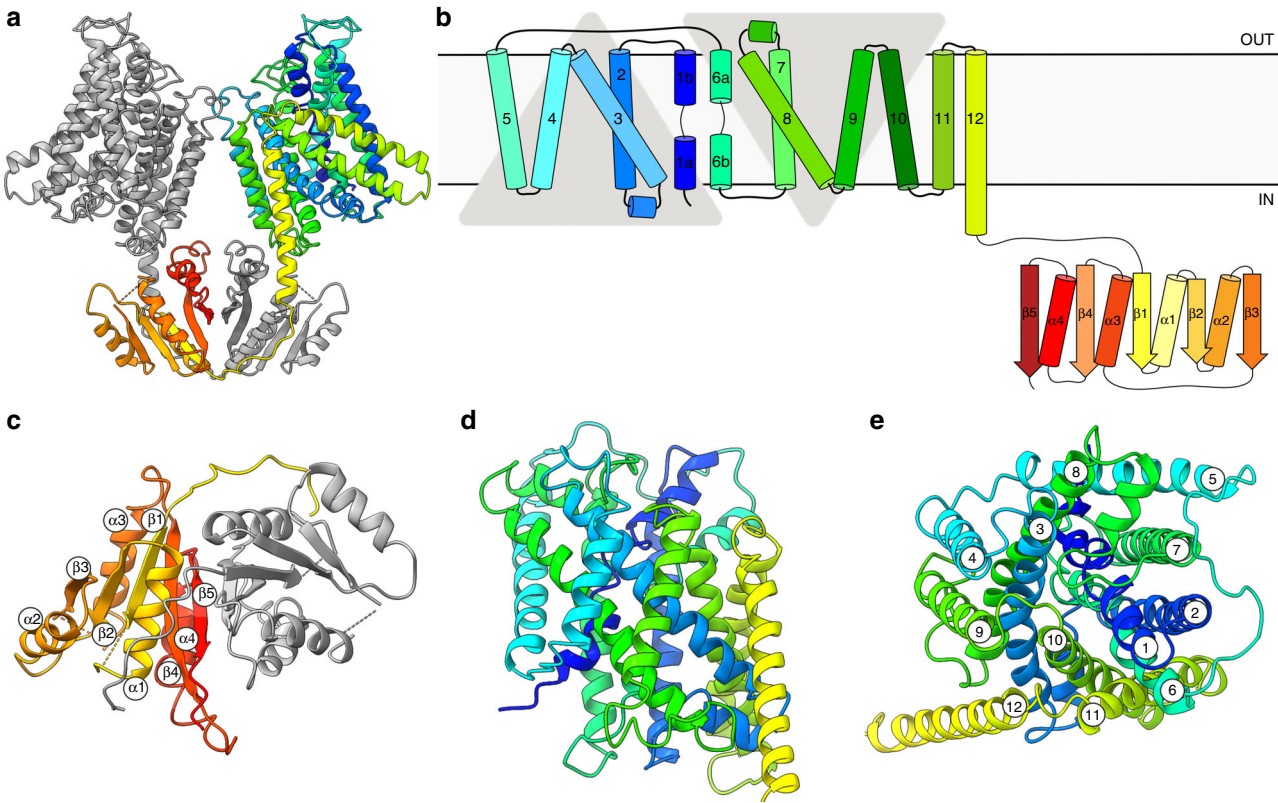

**Fig. 2 Overview of the structure of KimA. a** KimA dimer viewed from the membrane, with one monomer coloured in grey and the other in a rainbow, from blue (N terminus) to red (C terminus). **b** Schematic representation of the topology of KimA. Each KimA monomer is composed of 12 TM helices, organized in a 5 + 5 inverted structural repeat and two additional, C-terminal TM helices located at the periphery of the dimer. The last TM helix extends into the cytoplasm and positions the soluble domain below the TM domain of the second monomer. **c** The cytoplasmic domain of a KimA monomer is formed by a parallel, five-stranded β-sheet sandwiched by four α-helices, forming a continuous 10-stranded β-sheet in the dimer. **d** Side view and **e** top view of the membrane domain of a KimA monomer.

approximately twice as many lipids bind in between the proto-mers in the upright dimer compared to the tilted dimer (Fig. 3d). We speculate that not enough lipids were present after purification to fill the inter-dimer space and stabilize the upright position of the TM domains. Importantly, the inter-subunit movement did not affect the conformation of the TM domains over the course of the performed simulations. It remains elusive whether this movement has any physiological relevance or functional consequences.

The cytoplasmic domain of each protomer consists of β-strand-α-helix motifs, which form a parallel 5-stranded β-sheet flanked by four helices and in the dimer generate an extended β-sheet with 10 strands (Fig. 2a, c). Interestingly, the same fold was previously observed in a number of proteins, including soluble phosphopantetheine adenylyltransferases (PPAT) (PDB 1GN8)[25], which bind ATP and ADP (Supplementary Fig. 7), and the cytoplasmic domain of a prokaryotic cation-chloride cotranspor-ter (PDB 3G40)[26]. In analogy to the nucleotide binding to PPAT, the cytoplasmic domains of KimA may provide binding sites for cyclic di-AMP, which was recently identified as an inhibitor for the potassium uptake by KimA[27]. However, although 50 μM c-di-AMP was added to purified KimA before sample freezing, no additional density for the molecule was observed, suggesting that it could not bind to the present state. In agreement to this assumption, no c-di-AMP binding could be determined using isothermal titration calorimetry and thermal shift assay (data not shown).

Each TM domain contains 12 TM helices (Fig. 2b, d, e), the first ten of which adopt a typical LeuT fold[28], with a topologically

inverted repeat of TM helices 1-5 and TM helices 6-10 and broken helices 1 and 6 (Supplementary Fig. 8). TM helix 12 extends into the cytoplasm (Fig. 2) and, as previously mentioned, connects to the cytoplasmic domain. The structure represents an inward-occluded conformation of KimA. The extracellular side is tightly sealed by main chain packing of TM helices 1b, 3, 6a, and 10, while on the cytoplasmic side we observe a wide solvent-filled tunnel, lined by polar residues (D117, Y118, E233, N237, T317, S320, Q324) (Fig. 4a, b). A thin gate[29], formed by the side chains of D36, T121, T230 and Y377, separates this tunnel from a smaller cavity located next to the discontinuous region of TM helices 1 and 6, which we hypothesize to be a potassium ion binding site (Fig. 4a, b).

**Each TM domain has three potassium ions bound.** In agreement with the hypothesized potassium ion binding site, the cryo-EM map shows a non-protein density in this region, between D36, Y43, T121, S125, T230, and Y377, consistent with the presence of a potassium ion (Fig. 4c, d, Supplementary Fig. 3f). This location corresponds to the substrate binding site in other transporters with the LeuT fold.

Below the thin gate two other non-protein densities, separated by ~3 Å, are observed near the side chains of D36, D117, and E233 (Fig. 4e and Supplementary Fig. 3g). As the buffer contained 100 mM potassium chloride, and smaller molecules like waters are only visible at a resolution better than 3 Å, we also assign these densities as potassium ions (Fig. 4c–e). Their close proximity may indicate that their positions are not occupied at the same time and the map densities have been averaged.

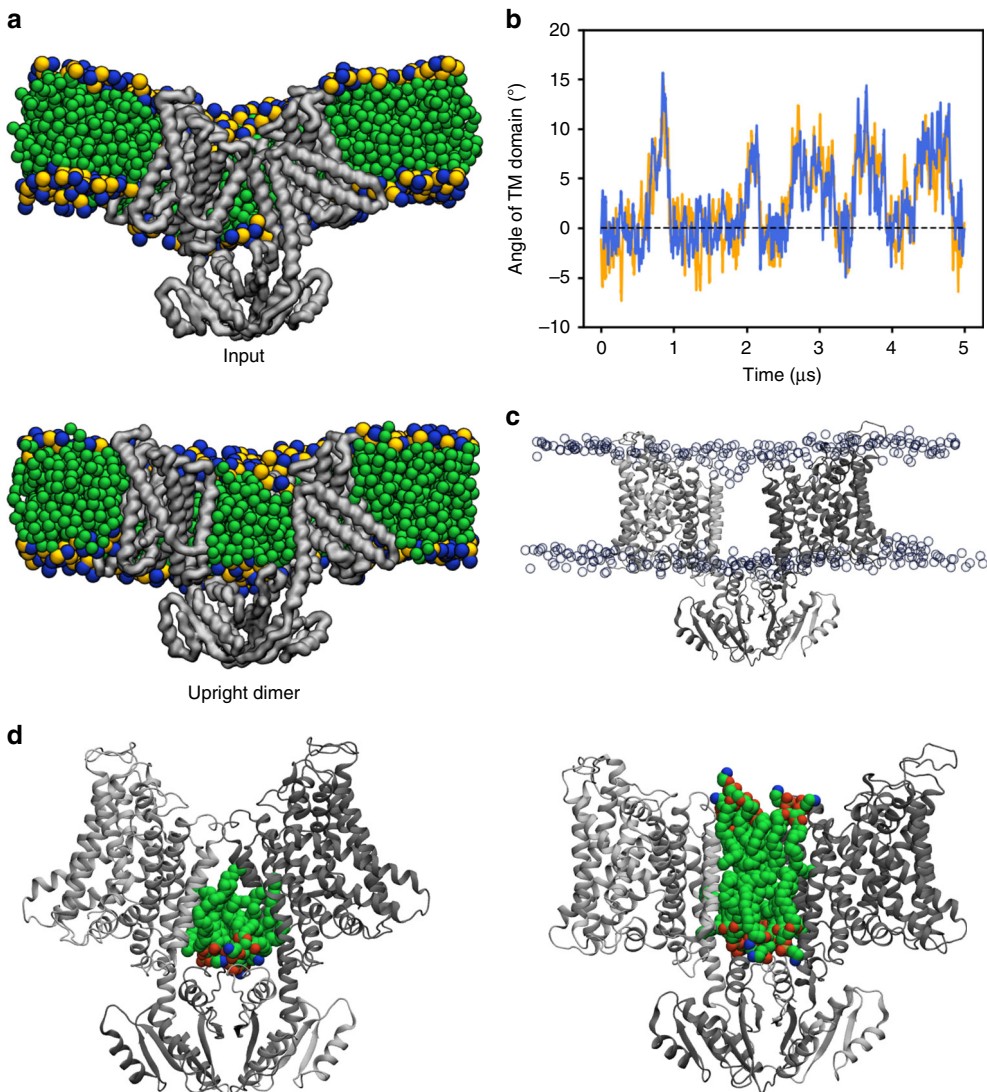

**Fig. 3 Molecular dynamics simulations of KimA in a lipid bilayer. a** Cut-away views of KimA dimer in a membrane, taken from the CG simulation data. The protein backbone is shown as grey surface and the POPE lipids as green, yellow and blue spheres. **b** Representative plot of the angle between the TM domains over time for the CG simulations; $n = 4$ independent simulations (see Supplementary Fig. 5 for full data and a schematic of how the angles were computed). **c** Upright dimer position as determined using ~2 µs of fully atomistic simulation. Protein shown as grey cartoon, lipid phosphates as semi-transparent blue circles. $n = 3$ repeats, with the upright dimer sampled in 2/3. **d** Views from atomistic simulation showing the lipid molecules (as coloured spheres) present between the KimA monomer.

Generally, the limited resolution of the map and possibly some dynamics of the ions led to comparably broad densities (Supplementary Fig 3f and g), which left some uncertainties for the assignment of correct potassium ion binding sites. Two 135 ns atomistic simulations of the dimer in 150 mM KCl and in the presence of the three bound $K^+$ ions were performed to detail the potassium ion binding. The simulations confirm the existence of three ion binding sites (Fig. 4f and Supplementary Fig. 6). On average, *ca.* 2.3 ions were bound to each monomer at any one time, with the ions dynamically sampling all three sites. Of the residues surrounding the occluded ion, a short minimum distance and high contact (residue-$K^+$ distance less than 0.4 nm for >75% of the simulation time) was observed with residues D36, T121 and Y377, while residues Y43 and T230 make either a more dynamic or more long-range contribution to $K^+$ coordination. A rather long minimum distance and hardly any contact was observed with S125. Hence, the occluded $K^+$ appears to be located close to the residues forming the thin, cytoplasmic gate, while S125, which was the closest residue to the density seen in

cryo-EM, is not directly involved in ion binding. The coordination of the two potassium ions below the thin gate was confirmed by the MD simulations, the minimum distances between the residues and the ions are similar to the distances determined in the structure. Strong contact was found with residues D36 and D117, while the interaction with E233 was dynamic; in both the cryo-EM map and the MD simulations, E233 is too far from the binding sites for an involvement in potassium ion coordination.

**Key residues for substrate binding and trans-inhibition**. To further evaluate the role of the residues in close proximity to the bound potassium ions, we tested point-mutated KimA variants for their ability to complement growth at $K^+$ limitation (Supplementary Figs. 9 and 10). For wild-type KimA a concentration of the half maximal growth ($K_S$) of 0.09 mM was determined. Of the residues surrounding the occluded ion, variants D36A/N and Y43A completely abolished $K^+$ uptake, while S125A, T121A and particularly Y377A led to reduced

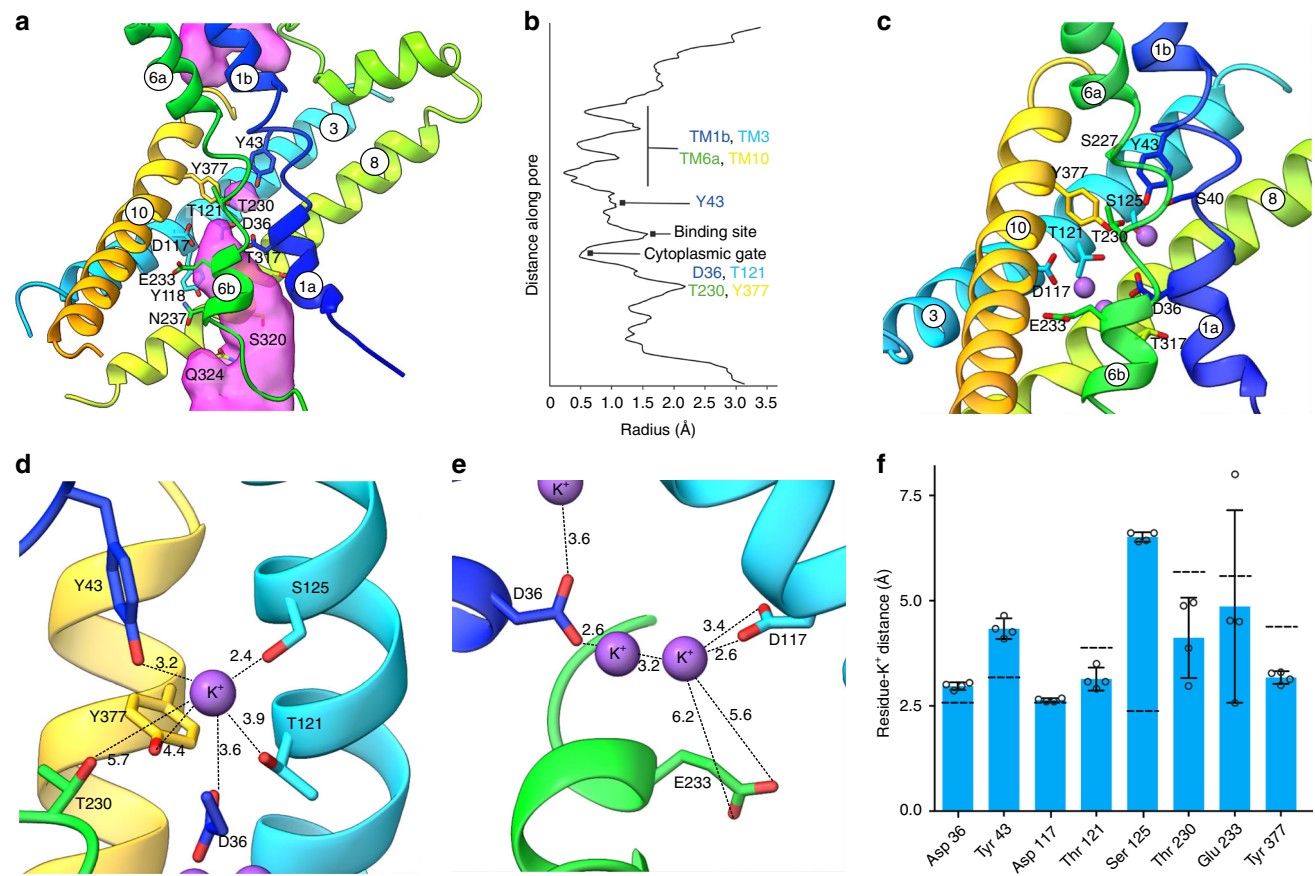

**Fig. 4 Structural details of the transmembrane domain. a** The translocation pathway reveals a tightly sealed extracellular side, a wide-open intracellular tunnel and the putative occluded potassium ion binding site, which is separated from the cytoplasm by a thin gate. The residues lining the intracellular entrance are represented in sticks. The surface representation of the pathway was calculated with HOLLOW[52]. **b** Pore radii along the translocation pathway of KimA determined by HOLE[53]. Features defining the inward-occluded state are highlighted. **c** Potential potassium ion binding sites within one protomer. **d**, **e** Close-up views of the potassium ions and the residues surrounding them with dotted lines showing the distances in Ångstrom. Potassium ions are depicted as purple spheres. **f** Average minimum distance for each residue to the closest $K^+$ ion over the course of two 135 ns atomistic simulations; errors shown are s.d., individual distances as dot plots, $n = 4$ repeats. Dashed lines indicate the distances derived from the structural model.

growth rates with $K_S$ values of 0.20, 0.27 and 11.0 mM, respectively. With a $K_S$ of 0.07 mM, variant T230A enabled a similar growth phenotype as the wild-type protein (Supplementary Fig. 9). The fact that some of these mutations did not simply abolish cell growth but showed a reduction of the apparent affinity corroborates their role in substrate binding and/or occlusion. The effects of mutating residues D36, T121, and Y377 are in agreement with a role in ion coordination. Y43 was suggested to only weakly interact with the potassium ion. Its localization towards the extracellular side of the TM domain suggests that it may instead form the extracellular gate of the binding site, which would explain the lethal effect of mutating it. Residue S125, which does not appear to be directly involved in potassium ion binding, may be important for the overall structure of the binding site and the high affinity binding, while residue T230 is not essential for potassium uptake.

Surprisingly, of the residues near the potassium ions in the cytoplasmic tunnel E233A/Q abolished $K^+$ uptake, while D117A/N/E had no effect on growth (Supplementary Fig. 8). Both the cryo-EM density and the MD simulations suggested that the essential residue E233 is not directly involved in coordinating the bound potassium ions (Fig. 4e, f). However, in the observed inward-facing conformation E233 has a predicted $pK_a$ of ~8 and thus could be easily protonated and deprotonated. For that reason, we hypothesized that E233 is responsible for proton coupling. Determining the $K^+/Rb^+$ exchange activity of $KimA_{E233A}$ confirmed this assumption: similar to wild-type KimA, the variant could still exchange $K^+$ for $Rb^+$, a process uncoupled from proton transport. In contrast, mutation D36A abolished $K^+/Rb^+$ exchange by KimA confirming the role of residue D36 in $K^+$ binding (Fig. 5a, b, Supplementary Fig. 11). Unlike E233, residue D117 clearly coordinates one of the potassium ions in the cytoplasmic tunnel, and it was therefore surprising that it is not required for transport. A possible explanation is that this residue acts as a sensor for the cytoplasmic potassium ion concentration, preventing further uptake by trans-inhibition. We propose that D117 may have a role in regulation by securing a fast response to variations in potassium concentration. In support of this hypothesis, mutation D117A led to diminished growth of *E. coli* LB2003 exclusively at high potassium concentration, while cells expressing wild-type KimA grew similar at all tested concentrations (Fig. 5c, d). The diminished growth at increased potassium concentrations argues for a toxic effect of an excess of potassium inside the cells as previously described[21,30]. The essential residue D36 coordinates the second potassium ion in the cytoplasmic tunnel as well as the potassium in the actual substrate binding site, and contributes to the thin gate. D36 appears to play a central role in KimA, participating not only in substrate binding and in gating, but also in regulation in a similar way to D117.

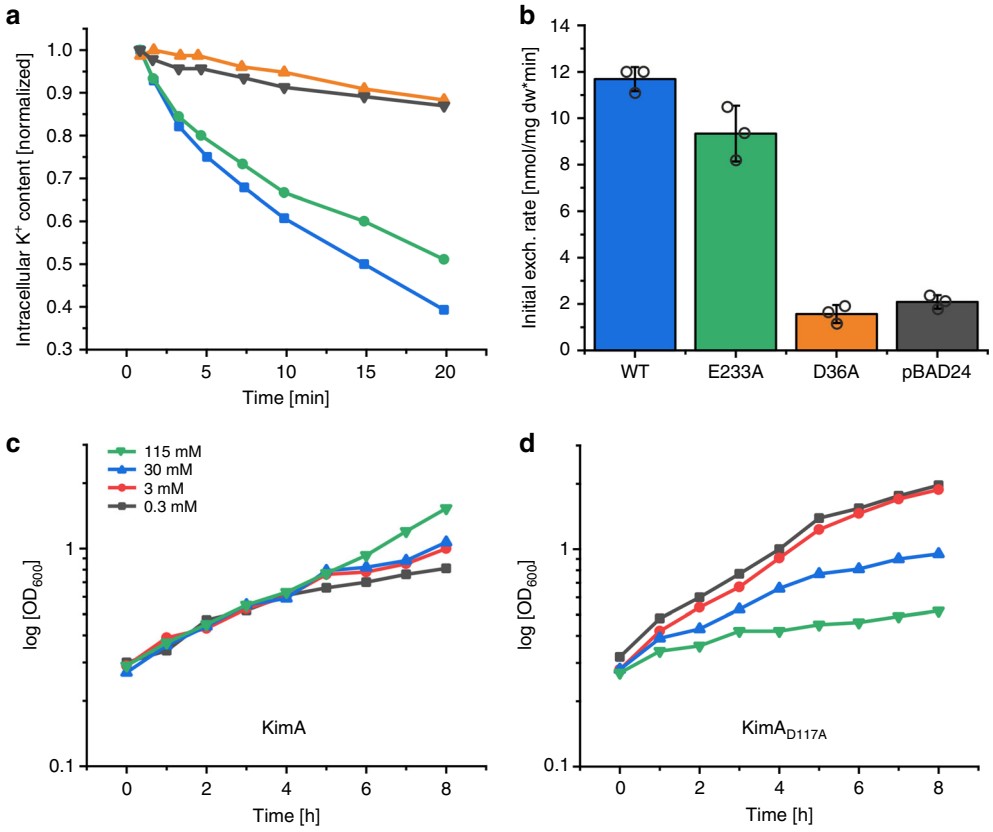

**Fig. 5 Involvement of residues in proton coupling or potassium binding or trans-inhibition. a** Time courses of the $K^+/Rb^+$ exchanges of *E. coli* LB2003 cells transformed with plasmids encoding wild-type KimA (blue), $KimA_{E233A}$ (green), $KimA_{D36A}$ (orange) or with the empty vector pBAD24 (black). To initiate the exchange, cells loaded with 50 mM KCl were diluted into an equivalent buffer with 50 mM RbCl. $n = 3$ independent experiments; a representative experiment is shown. **b** Initial rates of $K^+/Rb^+$ exchange given as means of three independent experiments; errors shown are s.d., individual exchange rates as dot plots. **c**, **d** Growth curves of *E. coli* LB2003 expressing wild-type KimA (**c**) and $KimA_{D117A}$ (**d**) at different potassium concentrations, ranging from 0.3 to 115 mM. $n = 3$ independent experiments; a representative experiment is shown. Source data are provided as a Source Data file.

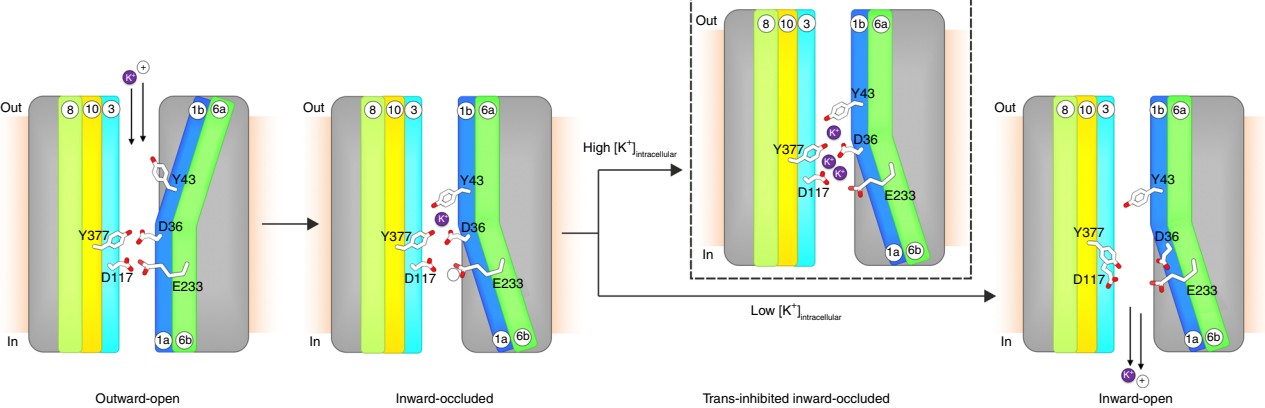

**Fig. 6 Schematic representation of alternating access transport in KimA.** The outward-open conformation should provide access of protons and potassium ions from the extracellular side to the binding sites, while at the cytoplasmic side the exit tunnel should be tightly sealed. The binding of a proton and a potassium ion could result in the movement of Y43, which serves as external gate occluding the bound substrate. Subsequently, broken helices TM1 and TM6 could alternate, sealing the extracellular tunnel and opening the intracellular tunnel. A thin, intracellular gate formed by residues D36 and Y377 could yet sustain an occluded state. Upon deprotonation of E233, the intracellular gate could open, allowing potassium ion release to the cytoplasm. At high intracellular potassium concentrations, potassium ions could bind within the intracellular tunnel preventing the opening of the intracellular gate. We hypothesize that our cryo-EM structure represents such a trans-inhibited, inward-occluded state (highlighted with dashed box).

## Discussion

Based on the presented data and the knowledge from other APC transporters[31,32], we propose a model for proton-coupled potassium transport via KimA (Fig. 6). To allow potassium and proton binding from the outside, KimA needs to adopt an outward-open conformation. Therefore, broken helices TM1a and TM6b probably move towards TM helices 3 and 10, tightly sealing the potassium ion binding site from the cytoplasm by main chain

packing. At the same time, orienting TM1b and TM6a away from TM helices 3 and 10 would open a tunnel to the extracellular space. To completely expose the binding site to the outside, residue Y43 probably reorients, functioning as the extracellular gate. Upon binding of at least one proton and one potassium ion to the respective binding sites (E233 for H$^+$, D36, T121 and Y377 for K$^+$), Y43 could occlude the bound ions from the extracellular side. Subsequently, TM1b and TM6a likely move towards TM helices 3 and 10 tightly sealing the extracellular side, while a displacement of TM1a and TM6b would open the intracellular tunnel. As seen in the presented structure, the potassium ion probably remains occluded towards the intracellular side by the thin intracellular gate formed by D36 and Y377. Ultimately, E233 could get deprotonated and trigger the opening of the intracellular gate, which would expose the binding site to the cytoplasm and allow potassium ion release. We suggest that the here-presented cryo-EM structure represents a trans-inhibited, inward-occluded state of KimA, in which the intracellular tunnel is filled with potassium ions at high internal potassium concentrations. We propose that these ions prevent the opening of the intracellular gate and the exposure of the potassium ion binding site towards the cytoplasm, hindering the further accumulation of potassium ions despite the presence of a driving proton gradient.

By performing a model-based sequence alignment of KimA with members of the KUP family, we found that the residues we identified as playing an important functional role in KimA are all fully or highly conserved among KUP members, while the sequence conservation in the KUP family is generally low (Supplementary Fig. 12). The overall structure of KimA is in agreement with the predicted overall structure of the low-affinity K$^+$ transporter Kup from *E. coli*, the best characterized KUP protein[11]. In addition, residues of *E. coli* Kup essential for transport, D23, E116 and E229[11] are equivalent to D36, D117, and E233 in KimA (Supplementary Fig. 12). However, the otherwise highly conserved VFGD/IYGD motif within TM helix 1 of KUP members is not conserved in KimA. This leads us to conclude that KimA actually forms a subfamily within the KUP family, and that the presented structure is relevant for all potassium transporters in this family. Our analysis provides the molecular basis for proton-coupled potassium ion transport in the KUP family and sets the framework for the full characterization of this widely distributed group of transporters, in particular the elucidation of the structural basis for the different potassium ion binding affinities within this family.

## Methods

**Cloning, expression and purification of KimA.** The gene encoding KimA was amplified from *B. subtilis* genomic DNA and cloned into a pBAD24 expression vector including a C-terminal His$_{10}$-tag. Point mutations in the *kimA* gene were applied by restriction-free cloning techniques. A list of all primers used is included in the Supplementary Methods. Wild-type KimA was produced in *E. coli* strain LB2003 (available from the authors upon request) by growing an overnight culture in KML medium supplemented with 100 µg/mL ampicillin. This culture was diluted 100-fold in 12 L of fresh KML medium, and overexpression was induced at an OD$_{600}$ of 0.8 by the addition of 0.002% arabinose. After 1.5 h of induction the cells were harvested and resuspended in 50 mM Tris pH 8, 100 mM KCl, 400 mM NaCl, 1 mM EDTA supplemented with 0.5 mM PMSF, 1 mM benzamidine and DNase I. The suspension was passed through a homogenizer at a pressure of 1 kbar to disrupt the cells. The cell extract was centrifuged at 15,000 × *g* for 15 min to remove unbroken cells and debris. Membranes were collected by centrifugation at 180,000 × *g* for 3 h. The membrane pellet was resuspended in solubilization buffer (50 mM Tris pH 8, 100 mM KCl, 400 mM NaCl) to 50 mg/mL and solubilized by the addition of 2% SMA co-polymer Xiran SL30010 P20 (2:1 molar ratio of styrene: maleic acid), provided by Polyscope Polymers B. V., at 4 °C overnight. Subsequently, the solution was centrifuged at 180,000 × *g* for 30 min to remove unsolubilized particles. The supernatant was incubated with Ni-NTA overnight. Then, the resin was washed with 50 column volumes of solubilization buffer supplemented with 50 mM imidazole to remove any unspecific protein. Finally, the protein was eluted using 500 mM imidazole in the solubilization buffer, and further purified by size exclusion chromatography using a Superose 6 10/300 GL column

(GE Healthcare) previously equilibrated with 50 mM Tris pH 8, 100 mM KCl. Fractions containing the protein were pooled and the sample was concentrated to 1 mg/mL prior to cryo-grid preparation. In the case of the detergent-solubilized KimA, the membrane pellet was solubilized with 1% DDM (from 20% stock). For the rest of that purification the buffers were supplemented with 0.04% DDM.

**K$^+$ uptake assay.** An adapted protocol from ref. [33] was established for potassium uptake experiments into intact, potassium-depleted *E. coli* LB2003 cells, which lack all endogenous potassium uptake systems. LB2003 transformed with the plasmid that encodes the protein of interest was grown in 1 mL KML medium supplemented with 100 µg/mL ampicillin at 37 °C shaking at 180 rpm overday. The preculture was used to inoculate 50 mL K30 medium (the number indicates the mM potassium concentration added to the minimal media) supplemented with ampicillin for an overnight culture. The next day, the cells were diluted to an OD$_{600}$ of 0.15 in 500 mL K30 medium containing ampicillin and 0.002% L-arabinose for gene expression. After the cells reached an OD$_{600}$ of ~0.6-0.8, they were centrifuged at 5000 × *g* at 15 °C for 10 min and the cell pellets were resuspended in 120 mM Tris-HCl pH 8 to an OD$_{600}$ of 30 and washed three times by centrifuging at 4000 × *g* at 20 °C for 10 min. After the last centrifugation step, the cells were adjusted to an OD$_{600}$ of 30 with the same buffer. The resuspended cells were incubated at 37 °C for 5 min. To release K$^+$ and Na$^+$ from the cytoplasm, the cells were treated with 1 mM EDTA while gently shaking at 37 °C for another 5 min. Subsequently, the cells were centrifuged twice to wash out EDTA and ions at 4000 × *g* at 20 °C for 8 min and the pellets were resuspended each time in 100 mM HEPES, 100 mM MES pH 7.5.

For the K$^+$ uptake experiment, cells were diluted into flasks to an OD$_{600}$ of 3 in 100 mM HEPES, 100 mM MES at the desired pH values under constant shaking at 20 °C. Ten minutes before initiating the potassium uptake, the cells were energized with 0.2% glycerol and 0.002% L-arabinose. The uptake was initiated by addition of various K$^+$ concentrations (0.05; 0.1; 0.2; 0.5; 1 and 2 mM). At different time points samples of 1 mL from each flask were transferred onto 200 µL silicone oil with a density of 1.04 g/cm$^3$ in a 1.5 mL centrifuge tube. The samples were immediately centrifuged at 17,000 × *g* at 20 °C for 2 min to isolate the cells from the supernatant. The supernatant and the silicon oil were removed and the cell pellets were cut with a razor blade and were added to 1 mL of 5% trichloracetic acid (TCA) solution. Cell pellets were resuspended by vortexing. The suspensions were frozen at −20 °C and subsequently cook at 90 °C for 10 min to release the cations. Afterwards the solutions were diluted with 3 mL 6.7 mM CsCl and 4 mL 5 mM CsCl. After removing the cell fragments by centrifugation at 4000 × *g* for 20 min the K$^+$ concentration was determined by flame photometry.

**K$^+$/Rb$^+$ exchange assay.** *E. coli* LB2003 cells were transformed with plasmids that encode KimA, KimA$_{D36A}$ and KimA$_{E233A}$, respectively, or with the empty vector pBAD24, and grown in K30 minimal medium as indicated above for the K$^+$ uptake assay. After harvesting, the cell pellets were resuspended and washed twice with 200 mM HEPES-TEA pH 7.5. Afterwards, cells were adjusted to an OD$_{600}$ of 300 with the same buffer supplemented with 50 mM KCl and incubated at room temperature for 3 h to load the cells with potassium ions. To initiate the exchange, cells were diluted 100-fold in 200 mM HEPES-TEA pH 7.5 supplemented with 50 mM RbCl (or 50 mM NaCl for the negative control) and with 30 µM of the proton ionophore carbonyl cyanide m-chlorophenyl hydrazone (CCCP). CCCP abolished the proton motive force, avoiding active transport. To monitor K$^+$/Rb$^+$ exchange, samples were taken at different time points, cells were treated as described for the K$^+$ uptake assay and intracellular K$^+$ concentrations were determined using flame photometry.

**Complementation assay.** Cells harbouring plasmids encoding for KimA and variants thereof were picked from agar KML plates and incubated in 1 mL KML medium supplemented with 100 µg/mL ampicillin at 37 °C shaking at 180 rpm overday. Then, the cells were incubated in 5 mL K30 medium supplemented with 100 µg/mL ampicillin, at 37 °C shaking at 180 rpm overnight. The next day, precultures were washed and centrifuged twice at 4000 × *g* for 10 min in K0 medium to remove external potassium. After that, 5 mL of different minimal media (K0.01, K0.02, K0.03, K0.05, K0.1, K0.2, K0.5, K1, K3, K10, K30 and K115) each supplemented with 100 µg/mL ampicillin and in the presence of 0.002% arabinose for gene expression was inoculated to an OD$_{600}$ of 0.15. The OD$_{600}$ was measured hourly for the first 10 h and after 24 h. After 24 h, samples were taken for SDS-PAGE and subsequent Western blotting to analyze the protein production.

**Preparation of proteoliposomes.** Proteoliposomes were prepared with a standard protocol[34]. Briefly, *E. coli* polar lipids in chloroform were dried in a rotary evaporator, resuspended to 10 mg/mL and sonicated in a buffer containing 100 mM NaP$_i$ pH 7. After three freeze-thaw cycles, large unilamellar vesicles were prepared by extrusion through a 400-nm diameter polycarbonate filter. Liposomes were diluted to 4 mg/mL and destabilized beyond R$_{sat}$ with Triton X-100. DDM-solubilized KimA was added to the liposomes at a weight ratio of 1:50 (protein:lipid), and detergent was subsequently removed by the addition of BioBeads. Proteoliposomes were harvested by centrifugation at 250,000 × *g* and resuspended to a lipid concentration of 10 mg/mL. After three freeze-thaw cycles, proteoliposomes were stored in liquid nitrogen until

use. For the preparation of the empty control liposomes SEC buffer was added to the lipids in the same ratio as the protein.

**Fluorescence-based transport assay**. Potassium-dependent proton transport was measured with the ΔpH-sensitive fluorophore 9-amino-6-chloro-2-methox-yacridine (ACMA). Proteoliposomes and empty liposomes, respectively, were thawed, extruded through a 400-nm polycarbonate filter, centrifuged and resuspended to a final concentration of 10 mg/mL lipids in 100 mM NaP$_i$ pH 7. For the fluorescence measurements, proteoliposomes and empty liposomes were diluted 100-fold into 100 mM KP$_i$ pH 7 and 500 nM ACMA. Fluorescence was recorded at an excitation wavelength of 410 nm and emission wavelength of 490 nm in a spectrofluorimeter over time. The K$^+$-coupled H$^+$ transport was initiated upon the addition of 500 nM sodium ionophore IV, which initially resulted in a membrane potential of −120 mV and hindered the establishment of an inhibitory positive potential. At the end of each experiment, the proton ionophore CCCP was added to dissipate the membrane potential allowing protein-independent proton fluxes.

**Sequence alignment**. Sequence conservation of KimA was evaluated by performing an alignment in T-coffee[35] using a list of genes obtained from the BLAST-NCBI server. Due to the low sequence similarity between KimA and members of the KUP family, a model-based sequence alignment between them was performed using the PROMALS3D[36] server. In general, the sequence alignment is a progressive method using database searches, secondary structure predictions, and available or provided 3D structures. For the presented alignments only database searches and secondary structure predictions were applied. The figures were prepared with Jalview[37].

**Negative-staining EM**. Samples of KimA solubilized with DDM or SMA co-polymer were negatively stained for up to 3 min with 1% (w/v) uranyl acetate pH 4. Electron micrographs were acquired with a CCD camera (Gatan Ultrascan 4000) on a Tecnai Spirit at 120 kV under low-dose conditions, at a magnification of 52,000× for a pixel size at the specimen of 2.11 Å. Particles were picked first manually with EMAN boxer[38] and, after generation of appropriate templates by 2D classification in Relion 2.1[39], automatically using Gautomatch (by Kai Zhang, MRC-LMB). A total of 3110 and 67,292 good particles of KimA in SMA lipid particles (SMALPs) and DDM, respectively, were picked. Two-dimensional class averages were obtained by 2D classification with Relion 2.1.

**Cryo-EM specimen preparation and data collection**. Cryo-EM grids of KimA in SMALP at 1.1 mg/mL with 50 μM cyclic di-AMP were prepared in a FEI Vitrobot plunge freezer at 10 °C and 90% humidity, using Quantifoil R2/2 holey carbon grids (Quantifoil Micro Tools), pre-treated in chloroform for 1 to 2 h and freshly glow-discharged. The grid was blotted for 9 sec and plunge-frozen in liquid ethane. Images were collected automatically using EPU (Thermo Scientific), on a FEI Titan Krios operating at 300 kV and aligned as previously described[40], with a Gatan K2 camera in counting mode and with an energy filter. The nominal magnification of 130,000x yielded a pixel size at the specimen of 1.077 Å. Each micrograph was recorded as a movie stack with 40 frames over 8 sec, with a calibrated dose of ~1.77 e$^-$/Å$^2$ per frame and defocus values between −0.5 and −3.2 μm.

**Image processing**. A set of 5418 micrographs was collected automatically, of which 4951 were of sufficient quality for processing. Drift correction and dose weighting of each movie stack were performed with MotionCor2[41]. Whole-micrograph CTF was determined with CTFFIND4[42] on drift-corrected, non-dose weighted movies. After manual picking of a small particle set using EMAN boxer[38], templates were generated (first by 2D classification and later by reprojection of a low-resolution 3D map) for automatic picking by template matching using Gautomatch (by Kai Zhang, MRC-LMB). The initial dataset contained 2,043,209 particles, windowed with a 208 pixels squared box. A low-resolution initial model was generated from ~19,000 particles using the stochastic gradient descent method implemented in Relion 2.1 and low-pass filtered to 60 Å[43].

A reference-free 2D classification with ISAC, within Sphire[44], was used to discard clear false positives and bad particles, outputting a subset with 1,614,900 particle images. A more homogeneous set of 314,399 particles was identified through two consecutive rounds of 3D classification, with five classes and no symmetry applied, in Relion 3.0[43]. After Bayesian polishing, CTF refinement and beam tilt estimation with Relion 3.0, refinement of this subset with a soft mask and applied C2 symmetry produced a map at 3.8 Å. However, the resolution degraded towards the periphery of the dimer, especially at the cytoplasmic domain, due to small variations in the relative position of the KimA monomers. Since the signal from a single monomer was insufficient for proper alignment, we attempted to identify a particle subset where these deviations were minimized through a 3D classification of the symmetry-expanded dataset (629,798 particles) with three classes, only local searches and a reference map low-pass filtered to 4.5 Å; after every five iterations, the map of each class was aligned to a partial map, composed of one transmembrane domain and one cytoplasmic domain, within Chimera[45], in order to keep the position of one half of the dimer constant, while the position of the other half drove the classification (a similar approach was implemented in

ref. [46]). The best 3D class contained 198,366 particles. Symmetry expansion was reversed by removal of duplicates, for a total of 149,724 unique particle images. A homogeneous refinement of these particles with cryoSPARC v2[47] produced a map with a nominal resolution of 3.7 Å and improved densities at the periphery. Local resolution was estimated with cryoSPARC v2[47].

**Model building and validation**. An homology model of the transmembrane domain of KimA obtained by the Phyre2 server[48] was docked initially into the cryo-EM map using USCF Chimera[45] and used as a starting point for modelling with Coot[49]. The cytoplasmic domain was built de novo. The model was then subjected to an iterative process of real space refinement using Phenix.real_space_refinement[50] with geometry and secondary structure restraints followed by manual inspection and adjustments in Coot[49]. The final model includes 572 residues of the 607 that compose a KimA monomer, lacking the first 26 amino acid residues at the N terminus and residues 480-481, 534-538 and 607 in the cytoplasmic domain. The geometries of the atomic model were evaluated by Mol-Probity[51]. The translocation pathway and the surface representation were obtained with HOLLOW[52]. Pore radii along the translocation pathway were calculated using HOLE software[53]. pK$_a$ calculations were performed with the multiconformation continuum electrostatics (MCCE) program[54]. Figures were prepared with UCSF ChimeraX[55] and PyMOL.

**CG simulations**. CG simulations of dimeric KimA were run using the CG Martini forcefield[23,24], using the open beta 3.0.b.3.2 version. Additional bonds of 500 kJ mol$^{-1}$ nm$^{-2}$ were applied between all protein backbone beads within 1 nm, except for interactions across the extracellular interface of the TM domains, designed to allow inter-domain dynamics and not bias the simulations towards the cryo-EM pose.

Initial simulations were run following the MemProtMD protocol[56,57]. The input protein was aligned accordingly on the *xy* plane, and POPE (1-palmitoyl-2-oleoyl-sn-glycero-3-phosphoethanolamine) lipids were placed randomly around the transmembrane region of protein, in a *z* range of 8 nm. All systems were solvated with Martini waters and Na$^+$ and Cl$^-$ ions to a neutral charge and 0.15 M. Self-assembly simulations were run to allow the membrane lipids to assemble into a bilayer around the protein. These were minimized using the steepest descents methods, then run over 100 ns in the NPT ensemble, with V-rescale temperature coupling[58] at 323 K and semi-isotropic Berendsen pressure coupling at 1 bar[59], with positional restraints of 1000 kJ mol$^{-1}$ nm$^{-2}$ applied to the backbone beads.

Following this, the systems were simulated for a further 4 × 5 μs, using V-rescale temperature coupling at 323 K and a semi-isotropic Parrinello-Rahman barostat[60]. Electrostatics were described using the reaction field method, with a cut-off of 1.1 nm using a potential shift modifier, and van der Waals interactions were shifted from 0.9-1.1 nm. Bonds were constrained using the LINCS algorithm. All simulations were run using Gromacs 2019[61]. Data were analyzed in Gromacs[62] or VMD[63], as described in the figure legends.

**Atomistic simulation**. Snapshots of the KimA dimer following 100 ns of restrained CG simulation in a POPC bilayer were converted to the CHARMM36 forcefield[64] following the CG2AT protocol[65]. Systems were solvated with TIP3P water and Na$^+$ and Cl$^-$ ions to 0.15 M. Electrostatics were handled using the Particle-Mesh-Ewald method, and a force-switch modifier was applied to the Van der Waals forces. Dispersion corrections were turned off. Systems were equilibrated for 2.5 ns with protein backbone restraints, before production simulations were run with V-rescale temperature coupling at 310 K using a time constant of 0.1 ps and Parrinello-Rahman semi-isotropic pressure coupling of 1 bar with a time constant of 2 ps, using 4 fs time steps with virtual-sites on the protein and lipids[66]. All simulations were run in Gromacs 2019[62].

For analyses of dimer stability, three simulations were run for ca. 2 μs to allow for the slower kinetics of dimer rearrangement to occur. For K$^+$ ion binding analysis, the K$^+$ ions of the structural model were reintroduced to the binding sites, and the system was re-solvated with K$^+$ and Cl$^-$ ions to 0.15 M. The bound ions and protein backbone were restrained for 1 ns of equilibration, before two production simulations of 135 ns were run, for a total of n = 4 ion binding sites.

**Reporting summary**. Further information on research design is available in the Nature Research Reporting Summary linked to this article.

## Data availability

Data supporting the findings of this manuscript are available from the corresponding authors upon reasonable request. A reporting summary for this Article is available as a Supplementary Information file. The source data underlying Figs. 1 and 5 and Supplementary Figs. 9 and 11 are provided as a Source Data file. The cryo-EM map and the model were deposited in the wwPDB with accession codes EMD-10092 and 6S3K, respectively.

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

## Acknowledgements

We thank Werner Kühlbrandt for his support of the cryo-EM. Jörg Stülke is acknowledged for helpful discussions about KimA, Arne Möller for helpful discussions on image processing, Mark Sansom for fruitful discussions on the MD simulations and Ahmadreza Mehdipour for assistance with pK$_a$ calculations. We thank Irfan Alibay and Owen Vickery for research software engineering support and implementation of lipid virtual sites for MD. Polyscope Polymers B. V. is acknowledged for providing SMA co-polymer Xiran SL30010 P20. This work was supported by the Max Planck Society (J.V.) and by the German Research Foundation via the Cluster of Excellence Frankfurt (Macromolecular Complexes) (I.H.). P.J.S. and R.A.C. were funded by Wellcome [208361/Z/17/Z]. Research in P.J.S.'s lab is funded by the MRC (MR/S009213/1) and BBSRC (BB/P01948X/1, BB/R002517/1 BB/S003339/1). This project made use of time on ARCHER granted via the UK High-End Computing Consortium for Biomolecular Simulation, HECBioSim (http://hecbiosim.ac.uk), supported by EPSRC (grant no. EP/R029407/1), Athena at HPC Midlands+, which is supported by the EPSRC (EP/P020232/1), and the University of Warwick Scientific Computing Research Technology Platform.

## Author contributions

I.T. and I.H. conceived the project. I.T. cloned, expressed and purified KimA wild-type and variants. I.T. and I.H. designed the functional experiments. I.T., D.G., N.A. and V.M. performed the functional experiments. D.J.M. and J.S.S. optimized the high-resolution EM alignment and the data collection procedure. J.S.S. prepared the samples for cryo-EM analysis, and collected and processed the data. I.T. and J.S.S. built and validated the atomic model. R.A.C. and P.J.S. performed the MD simulations. I.T., J.S.S., R.A.C., P.J.S., J.V. and I.H. interpreted the data. I.T. and J.S.S. wrote the initial draft and I.T., J.S.S., R.A.C., P.J.S., J.V. and I.H. wrote the manuscript. P.J.S., J.V. and I.H. supervised work, and P.J.S. and I.H. acquired funding.

## Competing interests

The authors declare no competing interests.
