## [Peer Review File · Nature Communications]

Reviewers' Comments:

Reviewer #1:

Remarks to the Author:

In the study, Tascón et al. report a 3.7 Å cryoEM structure of a potassium importer from *Bacillus subtilis* called KimA, which has been proposed to form a new family in the APC superfamily. They present functional evidence that KimA acts as a K⁺, H⁺ symporter. The cryoEM structure shows KimA as a dimer in an inward-occluded conformation. A number of interesting features are observed. The structure reveals two dimer interfaces formed by the crossing over of the cytoplasmic domain and tilting of the transmembrane domain which causes bending of the membrane. Three putative potassium ion densities were assigned and based on mutation-function correlation data, the authors propose residues involved in potassium binding and those involved in pH coupling and trans-inhibition. Finally, based on conservation of the functional residues in other KUP transporters which have also been proposed to function as K⁺, H⁺ symporters, it was concluded that KimA is a member of the KUP family.

Overall, the flux assays, cryoEM structure and MD simulation results support the conclusions. Given it is the first cryoEM structure solved in the KUP family, the study will be of interest to others in the field. However, some interpretations concerning the functional role of specific residues especially the proton coupling residue E233 and the trans-inhibition residue D117 are not well justified. The assignment of ion densities may not be accurate (note no cryoEM map was provided to the reviewer). This coupled with a lack of transport assays to directly test pH dependence or trans-inhibition make the claims of the functional roles of these residues highly speculative. Another issue is that although c-di-AMP was included in the sample, there was no mention in the results/discussion where the molecule is. This is disappointing as identification of the c-di-AMP binding site could offer insight into transport mechanism and its inhibition to increase the impact of the study.

Specific comments:

1. The authors have recently shown that c-di-AMP binds directly to KimA and inhibits K⁺ uptake. In the methods, it is stated that 50 μM of c-di-AMP was included in the sample. However, there is no specific discussion on the c-di-AMP binding site in the cryoEM density map (other than that the cytoplasmic domains may have a regulatory role as binding sites of cyclic di-AMP). This is an important point that needs to be addressed and seems to be one of the most interesting questions about this transporter.
2. According to CG MD simulation, the dimers fluctuate between two conformations with TMDs close together as seen in the cryoEM structure or separated. It is curious that only the inward occluded conformation is observed (or presented) in the paper. The authors should discuss this.
3. Fig.1c: The pH dependence experiment showed a range of pH from 6-8. At pH 8, there is still substantial K⁺ uptake. What is the intracellular pH of *E. Coli*? Is it expected that at pH 8, there is still significant proton gradient?
4. The inward occluded conformation shows tilted TM domains that seal the extracellular side. Is the breathing motion of the TMDs the mechanism for ion access to the transport tunnel?
5. Mutation of D117 did not have an effect on K⁺ transport. The authors propose that this residue may be involved in trans-inhibition. This seems speculative. A trivial explanation could be that the potassium ion density assigned is not correct. This also applies to the proposed role of E233 as a proton coupling residue and the regulatory role of D36. In the absence of definitive experimental evidence, the authors should tone down these statements and offer alternative interpretations
6. The title should be revised. It is unclear that this one structure provides understanding of the basis of proton-coupled K⁺ transport.
7. Supplementary Fig. 1 and 10: There is a half green half green star above an aspartate residue. What does this mean?

Reviewer #2:

Remarks to the Author:

The authors describe a combined functional and structural characterization of the bacterial KimA protein. The results provide novel insight into the structural architecture underlying proton/potassium transport in the KUP superfamily of transmembrane proteins. Overall, the paper is well written and interesting. However, the current manuscript leaves many questions unanswered, and would strongly benefit if these could be addressed.

First, the vital role of potassium homeostasis and the challenges for single-cell organisms is stressed as motivation for the current investigation. However, it appears that KimA is only involved in potassium uptake if other proteins known to play a role in potassium uptake are depleted. If this interpretation is correct, then it would seem that the role of KimA under physiological conditions (with those other channels not depleted) remains somewhat unclear. This should be discussed.

Second, this study provides what the authors refer to as an inward occluded state. In the overall transport cycle, is there any indication what role this state plays? Is it the resting state of the protein? Did the cryo-EM show any evidence of other highly populated states that might reflect other snapshots along the transport cycle?

Third, and somewhat related, the open dimer state as observed in the MD simulations is a relatively highly populated state under the simulation conditions. Currently, it remains somewhat unclear what role this state could play, but the presence of this state among the cryo-EM images would be a nice confirmation that this state is also sampled under the experimental conditions. Is there any state among the cryoEM classes that would be compatible with the open dimer state?

Fourth, it was mentioned that the ions were included in the simulations. How does their movement compare to the role of the involved residues discussed in the text on the basis of the cryoEM structure?

Finally, the proton transport mechanism remains largely undiscussed. It is only mentioned that E223, based on its unusual predicted pKa, may be involved in proton coupling. If it is indeed involved in proton translocation, then it should be expected to be part of some kind of proton cascade, perhaps not unlike e.g. bacteriorhodopsin. Are there nearby protonatable residues that could complete such a putative proton translocation pathway?

Reviewer #3:

Remarks to the Author:

The authors determined the detailed structure of *Bacillus subtilis* Kup, KimA transporter using cryo-EM. They formed homodimer. KimA had 12 transmembrane and showed Leu-T structure. The nucleotide binding sites and functional important negatively-charged residues in the transmembrane are confirmed by this structure. This structure provides important information on Kup transporters distributed in prokaryote and plant cells. The following comments may help improve the manuscript with more accurate information.

1 The title should be modified because some Kup did not show the proton-coupled potassium transport. The reports on direct measurement of Kup is limited. K uptake by *E. coli* Kup and *Arabidopsis* Kup was not dependent on pH dependency although there are reports on their proton coupling (which was not reproducible). One of possible title may be Structural basis of the proton-coupled potassium transport in *Bacillus subtilis* Kup.

2 The first paper on *E. coli* Kup by EP Bakker group should be cited with reference 2-3 because they named Kup from TrkD.

Bossemeyer, D., Schlosser, A., and Bakker, E.P. (1989) Specific cesium transport via the *Escherichia coli* Kup (TrkD) K⁺ uptake system. *J. Bacteriol.* 171, 2219-2221

3 The biochemical determination of 12 membrane spanning of *E. coli* Kup (Ref 10) are consistent with this data. This should be described in text.

4 The name of KimA does not represent their own structure and sometimes the name leads to confusing on their function. Therefore, it may be good to use KimA/Kup or rename Kup (KimA). Otherwise, readers can not correctly recognize this *B. subtilis* Kup correctly.

Reviewers' comments/authors' replay:

Reviewer #1 (Remarks to the Author):

In the study, Tascón et al. report a 3.7 Å cryoEM structure of a potassium importer from *Bacillus subtilis* called KimA, which has been proposed to form a new family in the APC superfamily. They present functional evidence that KimA acts as a K⁺, H⁺ symporter. The cryoEM structure shows KimA as a dimer in an inward-occluded conformation. A number of interesting features are observed. The structure reveals two dimer interfaces formed by the crossing over of the cytoplasmic domain and tilting of the transmembrane domain which causes bending of the membrane. Three putative potassium ion densities were assigned and based on mutation-function correlation data, the authors propose residues involved in potassium binding and those involved in pH coupling and trans-inhibition. Finally, based on conservation of the functional residues in other KUP transporters which have also been proposed to function as K⁺, H⁺ symporters, it was concluded that KimA is a member of the KUP family.

Overall, the flux assays, cryoEM structure and MD simulation results support the conclusions. Given it is the first cryoEM structure solved in the KUP family, the study will be of interest to others in the field. However, some interpretations concerning the functional role of specific residues especially the proton coupling residue E233 and the trans-inhibition residue D117 are not well justified. The assignment of ion densities may not be accurate (note no cryoEM map was provided to the reviewer). This coupled with a lack of transport assays to directly test pH dependence or trans-inhibition make the claims of the functional roles of these residues highly speculative. Another issue is that although c-di-AMP was included in the sample, there was no mention in the results/discussion where the molecule is. This is disappointing as identification of the c-di-AMP binding site could offer insight into transport mechanism and its inhibition to increase the impact of the study.

Specific comments:

1. The authors have recently shown that c-di-AMP binds directly to KimA and inhibits K⁺ uptake. In the methods, it is stated that 50µM of c-di-AMP was included in the sample. However, there is no specific discussion on the c-di-AMP binding site in the cryoEM density map (other than that the cytoplasmic domains may have a regulatory role as binding sites of cyclic di-AMP). This is an important point that needs to be addressed and seems to be one of the most interesting questions about this transporter.

We agree with the reviewer that the identification of the c-di-AMP binding site would be very exciting. Further, we expected that bound c-di-AMP could have decreased protein flexibility, which in turn could have increased the resolution of the cryo-EM map. Therefore, we included it in the cryo-EM sample, but unfortunately we could not observe any density that could be assigned to a c-di-AMP molecule. We speculate that c-di-AMP cannot bind to the present state of the protein. This was also supported by the fact that we could not determine any c-di-AMP binding using ITC and thermal shift assay. A specific comment was included in lines 125ff, which also further elaborates on the structural similarity between PPAT and the cytoplasmic domain of KimA. Additionally, a second panel has been added to supplementary figure 6.

2. According to CG MD simulation, the dimers fluctuate between two conformations with TMDs close together as seen in the cryoEM structure or separated. It is curious that only the inward occluded conformation is observed (or presented) in the paper. The authors should discuss this.

We thank the reviewer for highlighting this section of text. As it was previously written, we realise that the use of 'conformation' might have caused confusion with the transporter readership. In this instance 'conformation' does not relate to transport and the permeation of molecules through an individual KimA monomer. Instead it relates to the dimeric architecture, where we observe two distinct interactions at the oligomeric interface, which we term 'tilted' and 'upright'. In both of these two arrangements the individual transporter units remain in the same inward-occluded conformational state. It remains unclear whether the structural changes of the dimer interface have any physiological relevance. We further specify that the upright dimer is not observed among the molecules in cryo-EM. However, we do observe a degree of breathing motion in the cryo-EM data (lines 86ff). New atomistic MD simulations (lines 88ff.) lead us to suggest that not enough lipids were present to stabilize the upright form after purifying the protein.

3. Fig.1c: The pH dependence experiment showed a range of pH from 6-8. At pH 8, there is still substantial K⁺ uptake. What is the intracellular pH of *E. coli*? Is it expected that at pH 8, there is still significant proton gradient?

The intracellular pH of *E. coli* is around 7.5. Prior to the experiment, the cells were depleted of potassium ions. Hence, the observed K⁺ uptake at pH 8 is driven by the established potassium ion gradient and not by a proton gradient. In the revised manuscript we include a statement clarifying that KimA/Kup's major role is the K⁺ uptake at acidic pH (line 37).

4. The inward occluded conformation shows tilted TM domains that seal the extracellular side. Is the breathing motion of the TMDs the mechanism for ion access to the transport tunnel? As mentioned above (point 2) we apologize for the confusion. Despite the breathing motion at the dimer interface observed in the MD simulations, the protomers always adopted the inward-occluded conformation. The MD simulations were NOT designed to observe the transport mechanism but to evaluate whether the tilted dimer interface is reasonable. As mentioned in the revised manuscript (lines 97f.) it remains elusive whether the breathing motion has any physiological or functional relevance. For the transport mechanism we suggest a classical alternating access mechanism as described for other APC members. In the revised manuscript we have included a description of the envisioned transport mechanism (lines 199ff. and fig. 6).

5. Mutation of D117 did not have an effect on K⁺ transport. The authors propose that this residue may be involved in trans-inhibition. This seems speculative. A trivial explanation could be that the potassium ion density assigned is not correct. This also applies to the proposed role of E233 as a proton coupling residue and the regulatory role of D36. In the absence of definitive experimental evidence, the authors should tone down these statements and offer alternative interpretations

We thank the reviewer for raising this important point. We are confident that the ions are well assigned and therefore we provide the cryo-EM map and the pdb file for the reviewers. To provide more information on both the trans-inhibition and the proton coupling we have performed additional experiments for the revised manuscript.

A classical exchange experiment has been performed to prove the role of residue E233 in proton coupling. In 1988 Bossemeyer et al. described that *E. coli* Kup was able to transport Rb⁺ in the same fashion as K⁺. Therefore, we have developed an experiment, in which we have determined K⁺/Rb⁺ exchange by monitoring the efflux of potassium from cells loaded with K⁺ when diluted to a buffer with equal concentration of Rb⁺. CCCP was added to abolish any proton gradient, which otherwise could have supported active transport. In the assay

KimA_{E233A} supported K⁺/Rb⁺ exchange just like wild type KimA. By contrast, KimA_{D36A} was not able to exchange the ions. None of the tested variants could exchange K⁺ for Na⁺, demonstrating the specificity of ion transport. In summary, the exchange assay confirmed the suggested role of residue E233 in proton coupling and also the role of D36 in K⁺ binding (cf. lines 179ff., fig. 5a,b, and supplementary fig. 10).

To address the suggested trans-inhibition by K⁺ bound to D117, we have performed a growth complementation assay with *E. coli* LB2003 expressing wild type KimA and KimA_{D117A} at potassium concentrations ranging from 0.3 mM to 115 mM. In the presence of mutant KimA_{D117A}, we observed significantly reduce growth at 30 and 115 mM K⁺, which indicates a toxification of the cells by an excess of internal K⁺. In contrast, wild type KimA supported the identical growth at all tested conditions. The result clearly indicates that K⁺ binding to D117 hinders the uncontrolled uptake of potassium at elevated external potassium concentration, supporting the suggested trans-inhibition (cf. lines 188ff. and fig. 5c,d).

6. The title should be revised. It is unclear that this one structure provides understanding of the basis of proton-coupled K⁺ transport.

We really think that together with the new data on proton coupling and trans-inhibition, our paper does provide an understanding of proton-coupled K⁺ transport. To stress this further we have included a new fig. 6 with a model for transport (lines 199ff.). To address the criticism of reviewer 3, comment 1 that some KUP transporters may not be proton coupled, we slightly changed the title to „Structural basis of proton-coupled potassium transport in the KUP family”, leaving open the possibility that uncoupled potassium transporters exist within the KUP family.

7. Supplementary Fig. 1 and 10: There is a half green half green star above an aspartate residue. What does this mean?

The green star marks a residue involved in potassium binding in the intracellular tunnel and the purple stars indicate residues involved in potassium binding to the substrate binding site. Therefore, the half green, half purple star marks a residue coordinating potassium ions at both sites. We apologize for the confusion and we have corrected the figure legends in the updated manuscript.

Reviewer #2 (Remarks to the Author):

The authors describe a combined functional and structural characterization of the bacterial KimA protein. The results provide novel insight into the structural architecture underlying proton/potassium transport in the KUP superfamily of transmembrane proteins. Overall, the paper is well written and interesting. However, the current manuscript leaves many questions unanswered, and would strongly benefit if these could be addressed.

First, the vital role of potassium homeostasis and the challenges for single-cell organisms is stressed as motivation for the current investigation. However, it appears that KimA is only involved in potassium uptake if other proteins known to play a role in potassium uptake are depleted. If this interpretation is correct, then it would seem that the role of KimA under physiological conditions (with those other channels not depleted) remains somewhat unclear. This should be discussed.

We thank the reviewer for pointing out that we have not clarified the physiological role of KimA. KUP transporters are believed to be particularly important at acidic environmental conditions. We have included a comment on that in line 37. However, as it retains activity at neutral pH, KimA can also complement the lack of KtrAB in *B. subtilis*.

Second, this study provides what the authors refer to as an inward occluded state. In the overall transport cycle, is there any indication what role this state plays? Is it the resting state of the protein? Did the cryo-EM show any evidence of other highly populated states that might reflect other snapshots along the transport cycle?

Generally, the inward-occluded state precedes the opening of the intracellular gate upon deprotonation of the residue involved in proton coupling, which triggers the release of the potassium ion bound at the substrate binding site. We postulated that this residue is E233 and now provide further evidence with the exchange assay performed (cf. comment 5 to reviewer 1, lines 179ff., fig. 5a,b, and supplementary fig. 10). In general, most structures solved of APC members reflect the inward-occluded state, suggesting that it is highly populated and particularly stable. In our specific case, we believe that KimA is trapped in this conformation by the suggested trans-inhibition in the presence of high internal potassium concentrations. To support the trans-inhibition, we are providing new experimental data (cf. comment 5 to reviewer 1, lines 188ff., fig. 5c,d). In agreement to the suggested trapping, the cryo-EM data did not show any other state reflecting another conformation along the transport cycle.

Third, and somewhat related, the open dimer state as observed in the MD simulations is a relatively highly populated state under the simulation conditions. Currently, it remains somewhat unclear what role this state could play, but the presence of this state among the cryo-EM images would be a nice confirmation that this state is also sampled under the experimental conditions. Is there any state among the cryoEM classes that would be compatible with the open dimer state?

In cryo-EM, we did not observe classes with completely separated TM domains. However, the particles showed some flexibility that correlates with the breathing motion observed in the MD simulations (cf. lines 86ff.). Based on the now included MD simulations, we suggest that the upright dimer interface could actually not be adapted in the purified protein sample because not enough lipids were present to fill the larger gap between the protomers (lines 94ff., fig. 3e).

We think it is important to clarify that we actually performed the MD simulations because we were not sure whether the state we solved using cryo-EM would ever exist in vivo. Instead we

were concerned that a delipidation during purification might have led to the tilted dimer. Thus, the MD simulations give us confidence that, in principle, both states, the upright and the tilted dimer, could exist. With respect to the function of both states, it is important to mention that both show the identical inward-occluded conformation of the TM domains. It thus remains elusive whether the dimer interface is somehow involved in the regulation of activity; for example by effecting the binding of inhibitory cyclic di-AMP (cf. lines 95ff. and 125ff.)

Fourth, it was mentioned that the ions were included in the simulations. How does their movement compare to the role of the involved residues discussed in the text on the basis of the cryoEM structure?

We have not specifically looked at potassium ions within our simulations. As mentioned to reviewer 1, the MD simulations were designed to evaluate whether the tilted dimer interface is reasonable.

Finally, the proton transport mechanism remains largely undiscussed. It is only mentioned that E223, based on its unusual predicted pKa, may be involved in proton coupling. If it is indeed involved in proton translocation, then it should be expected to be part of some kind of proton cascade, perhaps not unlike e.g. bacteriorhodopsin. Are there nearby protonatable residues that could complete such a putative proton translocation pathway?

Please refer to the 5th point of reviewer 1 regarding the involvement of E233 in proton coupling. We are providing K⁺/Rb⁺ exchange experiments proving its function.

To better discuss the proton-coupled K⁺ transport by KUP transporters we are now providing a model to propose a possible transport mechanism (lines 199ff, fig. 6). Since residue E233 is already lining the intracellular tunnel we do not think there will be any protonisable residues downstream. To our knowledge, secondary active transporters normally do not present any kind of proton cascade. However, in some proton-coupled transporters proton sensors at the extracellular side have been identified that transfer the protons to the protonisable residues close to the substrate binding site. Since we could only solve the inward-occluded state, we could not identify any such sensors and it remains unclear whether there are any further proton-binding sites.

Reviewer #3 (Remarks to the Author):

The authors determined the detailed structure of Bacillus subtilis Kup, KimA transporter using cryo-EM. They formed homodimer. KimA had 12 transmembrane and showed Leu-T structure. The nucleotide binding sites and functional important negatively-charged residues in the transmembrane are confirmed by this structure. This structure provides important information on Kup transporters distributed in prokaryote and plant cells. The following comments may help improve the manuscript with more accurate information.

1 The title should be modified because some Kup did not show the proton-coupled potassium transport. The reports on direct measurement of Kup is limited. K uptake by E. coli Kup and Arabidopsis Kup was not dependent on pH dependency although there are reports on their proton coupling (which was not reproducible). One of possible title may be Structural basis of the proton-coupled potassium transport in Bacillus subtilis Kup.

We apologize for this misunderstanding. The title was not meant to necessarily say that all the potassium transporters in the KUP family are proton-coupled. It means that the presented structure and the functional data uncover the basis for proton-coupled transporters within the KUP family. That said, we are not entirely convinced there are some KUP members that are not proton coupled. In any case, we believe that our data and in particular the structure will help to further investigate the proton-coupling in other KUP members and ultimately clarify this uncertainty. We have changed the title to „**Structural basis of proton-coupled potassium transport in the KUP family**” leaving open the possibility that uncoupled potassium transporters exist.

2 The first paper on E. coli Kup by EP Bakker group should be cited with reference 2-3 because they named Kup from TrkD.

Bossemeyer, D., Schlosser, A., and Bakker, E.P. (1989) Specific cesium transport via the Escherichia coli Kup (TrkD) K⁺ uptake system. J. Bacteriol. 171, 2219-2221

We thank reviewer 3 for the comment. We have included the reference in the updated manuscript.

3 The biochemical determination of 12 membrane spanning of E. coli Kup (Ref 10) are consistent with this data. This should be described in text.

We have included a comment in the updated manuscript in lines 224ff.

4 The name of KimA does not represent their own structure and sometimes the name leads to confusing on their function. Therefore, it may be good to use KimA/Kup or rename Kup (KimA). Otherwise, readers can not correctly recognize this B. subtilis Kup correctly.

We have given seriously thought to the renaming of KimA, indeed even discussed it with Jörg Stülke, who first discovered the protein. As demonstrated in our manuscript, KimA clearly resembles the overall structure expected for all KUP members and the key residues involved in potassium transport appear to be highly conserved. Yet, KimA does not contain the highly conserved sequence motif of VFGD/IYGD in TM helix 1. Thus we suggest that KimA proteins form a new subfamily of KUP, a family that already unites Kup, HAK and KT. We have detailed our argument in the revised manuscript (lines 224ff.), while suggesting not to rename KimA.

Reviewers' Comments:

Reviewer #1:

Remarks to the Author:

The authors have done a good job addressing this reviewer's concerns. The additional experiments strengthen the conclusions on the functional roles of the specific residues. Below are minor issues that need to be addressed.

Fig.3: The label "e" in the figure should be corrected to "d" to match the legend.

Fig.5, Fig.S10: Statistical analyses should be provided.

Reviewer #2:

Remarks to the Author:

All my comments were satisfactorily addressed, except for the role of the ions in the MD simulations. It is a missed opportunity to not cross check the proposed ion behavior based on the cryoEM structure with the MD simulations. Even if the simulations were conducted for another purpose, the ion behavior is included there and therefore these trajectories provide a straightforward opportunity to investigate the proposed mechanism.

Reviewer #3:

Remarks to the Author:

The authors addressed several of the issues raised except for No. 3. A bit more care could have been taken before this work can be published; "Line38-39 due to the complete lack of structural information their transport mechanism remains unknown." In 2014, biochemical experimental approaches by alkaline phosphatase fusions and cysteine scanning have first determined the 12 spanning membrane structures of E. coli Kup (J. Biochem, 2014, Ref 11) and identified the acidic residues responsible for Kup-mediated uptake activity as their experimental information. The current manuscript by Tascón et al. contains cryo-EM-mediated structural analysis, which may be conclusive output and can be highly appreciated. Therefore, the evidence on 12 membrane-spanning of Kup homolog in E. coli (Ref 11) should be cited in Line 39 (Introduction) and Line 132 (or Line 224). Description of this fact will not lose the high value of this manuscript by Tascón et al.

Frankfurt, 4 December 2019

Dear Reviewers,

First of all we would like to thank you all for appreciating the improvement our manuscript “Structural basis of proton-coupled potassium transport in the KUP family“. We hope we now implemented all suggested changes, particularly the evaluation of the behavior of potassium ions using atomistic MD simulations. Please find a point-to-point discussion below.

Reviewers' comments/authors' replay:

Reviewer #1 (Remarks to the Author):

The authors have done a good job addressing this reviewer’s concerns. The additional experiments strengthen the conclusions on the functional roles of the specific residues. Below are minor issues that need to be addressed.

Fig.3: The label “e” in the figure should be corrected to “d” to match the legend.

Thanks for the notification. This has been corrected.

Fig.5, Fig.S10: Statistical analyses should be provided.

As for all the other functional data, a representative experiment is shown. A statistical analysis is included in the bar graphs; further the replicates are included in the provided source data file

Reviewer #2 (Remarks to the Author):

All my comments were satisfactorily addressed, except for the role of the ions in the MD simulations. It is a missed opportunity to not cross check the proposed ion behavior based on the cryoEM structure with the MD simulations. Even if the simulations were conducted for another purpose, the ion behavior is included there and therefore these trajectories provide a straightforward opportunity to investigate the proposed mechanism.

We apologize for not having addressed this point before but actually the previous simulations performed were in the absence of the potassium ions because they were not of importance for the point we wanted to address. However, we agree with the reviewer that MD simulations would add on the identification of the potassium ion binding sites and therefore we performed new atomistic simulations in the presence of the three potassium ions (lines 132ff.). The simulations nicely confirmed all three ion binding sites and further allowed the identification of residues that have a major contribution in ion binding.

Reviewer #3 (Remarks to the Author):

The authors addressed several of the issues raised except for No. 3. A bit more care could have been taken before this work can be published; "Line38-39 due to the complete lack of structural information their transport mechanism remains unknown." In 2014, biochemical experimental approaches by alkaline phosphatase fusions and cysteine scanning have first determined the 12 spanning membrane structures of E. coli Kup (J. Biochem, 2014, Ref 11) and identified the acidic residues responsible for Kup-mediated uptake activity as their experimental information. The current manuscript by Tascón et al. contains cryo-EM-mediated structural analysis, which may be conclusive output and can be highly appreciated. **Therefore, the evidence on 12 membrane-spanning of Kup homolog in E. coli (Ref 11) should be cited in Line 39 (Introduction) and Line 132 (or Line 224). Description of this fact will not lose the high value of this manuscript by Tascón et al.**

We apologize if you felt that we have not appreciated the previous work enough. We now have adapted the introduction already mentioning the predicted 12 TMs and referring to several paper (not just the one suggested) that have identified residues involved in potassium ion transport in the different KUP family members (lines 36ff.). Further, we already had included the mentioned paper in the last section of our main text highlighting that many of the authors' conclusions are now confirmed (lines 204ff.). We do not want to make this statement earlier in the text as suggested by the reviewer because only by the alignment discussed in this section we came to the conclusion that KimA is a KUP member.

Reviewers' Comments:

Reviewer #2:

Remarks to the Author:

It is reassuring to see that the authors have now taken the opportunity to compare the potassium ion distribution from simulation to the ion positions in the structure. However, I disagree with the notion of the authors that "The simulations nicely confirmed all three ion binding sites". As is visible from comparison of Fig. 4d and 4f, the coordination by S125 (the closest contact in the EM structure) is not confirmed by the simulation (zero contact). Rather than confirming the assigned binding sites, the simulation findings would thus seem to question the experimentally assigned binding sites. This is also seen in Fig. S6, where all of the experimentally assigned ion binding positions are outside the mesh of the MD based potassium densities.

minor: in Fig 4f contacts are reported as "%" but the numbers between zero and one suggest that they are fractions rather than percentages.

Reviewers' comments/authors' reply:

Reviewer #2 (Remarks to the Author):

It is reassuring to see that the authors have now taken the opportunity to compare the potassium ion distribution from simulation to the ion positions in the structure. However, I disagree with the notion of the authors that "The simulations nicely confirmed all three ion binding sites". As is visible from comparison of Fig. 4d and 4f, the coordination by S125 (the closest contact in the EM structure) is not confirmed by the simulation (zero contact). Rather than confirming the assigned binding sites, the simulation findings would thus seem to question the experimentally assigned binding sites. This is also seen in Fig. S6, where all of the experimentally assigned ion binding positions are outside the mesh of the MD based potassium densities.

Dear reviewer, first of all thank you for your fast response. We agree with you that the positions of the ions as allocated based on the map and the main positions found in the simulations are not exactly the same. What we intended to say was that the presence of three ion binding sites and their approximate location were confirmed by the simulations. The differences found most likely were caused by uncertainties in the 3.7-Å cryo-EM map. This map allowed us to identify the fold of the protein and enabled the observation of densities compatible with potassium ions. However, it was not good enough to unambiguously allocate the potassium ions. We clarified this by introducing the new MD simulations as follows:

"Generally, the limited resolution of the map and possibly some dynamics of the ions led to comparably broad densities (Supplementary Fig 3 f and g), which left some uncertainties for the assignment of correct potassium ion binding sites. Two 135 ns atomistic simulations of the dimer in 150 mM KCl and in the presence of the three bound K⁺ ions were performed to detail the potassium ion binding. The simulations confirm the existence of three ion binding sites..." (lines 130ff).

In the following lines (135ff.) we detail the similarities and differences between the model and the simulations, particularly mentioning the discrepancies seen for residue S125. To better highlight the general agreement between our model and the simulations we have also included a new panel f in figure 4, which instead of the contacts closer than 0.4 nm gives the mean distance of residues in the MD simulations to the closest potassium ion. You can see that for all residues except S125 these distances are in good agreement with the distances resulting from the cryo-EM data. The original panel 4f moved to figure S6.

Finally, we also included the results from the MD simulations in the interpretation of the subsequent mutational studies (lines 149ff.).

Minor: in Fig 4f contacts are reported as "%" but the numbers between zero and one suggest that they are fractions rather than percentages.

Thanks for the notification. This has been corrected.

Reviewers' Comments:

Reviewer #2:

Remarks to the Author:

The authors have satisfactorily addressed my concerns.